# Expression and Biological Functions of miRNAs in Chronic Pain: A Review on Human Studies

**DOI:** 10.3390/ijms23116016

**Published:** 2022-05-27

**Authors:** Saverio Sabina, Alessandra Panico, Pierpaolo Mincarone, Carlo Giacomo Leo, Sergio Garbarino, Tiziana Grassi, Francesco Bagordo, Antonella De Donno, Egeria Scoditti, Maria Rosaria Tumolo

**Affiliations:** 1Institute of Clinical Physiology, National Research Council, Via Monteroni, 73100 Lecce, Italy; sabina@ifc.cnr.it (S.S.); leo@ifc.cnr.it (C.G.L.); mariarosaria.tumolo@unisalento.it (M.R.T.); 2Department of Biological and Environmental Sciences and Technology, University of Salento, Via Monteroni, 73100 Lecce, Italy; alessandra.panico@unisalento.it (A.P.); tiziana.grassi@unisalento.it (T.G.); antonella.dedonno@unisalento.it (A.D.D.); 3Institute for Research on Population and Social Policies, National Research Council, c/o ex Osp. Di Summa, Piazza Di Summa, 72100 Brindisi, Italy; pierpaolo.mincarone@irpps.cnr.it; 4Department of Neuroscience, Rehabilitation, Ophthalmology, Genetics and Maternal/Child Sciences, University of Genoa, 16132 Genoa, Italy; sgarbarino.neuro@gmail.com; 5Department of Pharmacy-Pharmaceutical Science, University of Bari Aldo Moro, Via Edoardo Orabona, 70126 Bari, Italy; francesco.bagordo@uniba.it

**Keywords:** miRNA, miRNA target, chronic primary pain, chronic secondary pain, pain pathogenesis

## Abstract

Chronic pain is a major public health problem and an economic burden worldwide. However, its underlying pathological mechanisms remain unclear. MicroRNAs (miRNAs) are a class of small noncoding RNAs that post-transcriptionally regulate gene expression and serve key roles in physiological and pathological processes. This review aims to synthesize the human studies examining miRNA expression in the pathogenesis of chronic primary pain and chronic secondary pain. Additionally, to understand the potential pathophysiological impact of miRNAs in these conditions, an in silico analysis was performed to reveal the target genes and pathways involved in primary and secondary pain and their differential regulation in the different types of chronic pain. The findings, methodological issues and challenges of miRNA research in the pathophysiology of chronic pain are discussed. The available evidence suggests the potential role of miRNA in disease pathogenesis and possibly the pain process, eventually enabling this role to be exploited for pain monitoring and management.

## 1. Introduction

Chronic pain (CP), an unpleasant sensory and emotional experience that affects many persons worldwide and is a leading chief complaint in ambulatory care settings [1]. It is often the result of damage to and persistent inflammation of peripheral (somatic) or internal (visceral) non-neural tissue (nociceptive pain), or a lesion or disease of the central or peripheral somatosensory nervous system (neuropathic pain) [2]. The International Classification of Diseases (ICD-11) distinguishes two main categories of CP: chronic primary pain (CPP) and chronic secondary pain (CSP). The former persists or recurs for longer than 3 months where the pain can be conceived as a disease and cannot be explained by another chronic condition. CSP appears as a symptom of another disease but is a problem in its own right in many cases, persisting beyond the successful treatment of the initial causing pathology (Table 1) [3]. It is estimated that 20% of European adults have had CP [4]. A similar estimated prevalence has been reported in US adults [5].

CP conditions are identified as a leading cause of global disability [6] regardless of its cause, the body site and the age of the patients. In Europe, its economic burden, in terms of cost per patient per year, has been estimated as between 1800 euros and 10,200 euros, depending on the severity of the CP and the country; its economic weight and long course make it one of the most expensive long-term illnesses for the community [7,8]. CP causes adverse effects on psychological functions (depression, anxiety, coping skills, post-traumatic stress, etc.) as well as on sociocultural aspects (low educational attainment, poor social support, financial barriers or health insurance, substance abuse, etc.), and therefore has negative consequences for individuals, families and society [9].

Current evidence on the management of CP recommends an interdisciplinary approach, including restorative therapies, pharmacotherapies, procedural interventions, behavioral treatments, complementary therapies and integrative therapies [9]. Nonetheless, most patients do not receive specialist pain treatment, and 40% reported inadequate pain management [4]. The absence of optimal therapy is linked to the complexity of CP pathophysiology [10].

Accumulating evidence suggests that the field of epigenetics, i.e., changes in gene expression without modifying genomic sequences, may provide clues for CP management [9]. Among epigenetic mechanisms, microRNAs (miRNAs or miRs) are involved in physiological and pathological processes, and their alteration contributes to a variety of diseases [11,12]. MiRNAs are small noncoding single-stranded RNAs of approximately 22 nucleotides in length that participate in RNA silencing and post-transcriptional regulation of gene expression [13]. They play pivotal roles in many biological processes [14] and have been detected not only inside the cells, but also in extracellular human body fluids [15,16]. Extracellular miRNAs can be packed into apoptotic bodies, microvesicles, exosomes or bond to a protein, so they are highly stable despite the extracellular RNase activities [17]. Due to these properties, miRNAs could be a candidate biomarker in many physiologic and pathologic processes.

There are several studies reporting changes in miRNA expression in musculoskeletal disorders, migraines and other pain conditions. Moreover, a single miRNA can have multiple target genes and regulate many different biological pathways involved in pain processing [18]. Identifying the interactions that occur between miRNAs and their targets is a critical step in defining the regulatory role of miRNAs in the complex networks that regulate biological processes. Manually predicting the interaction of an miRNA with its target is not feasible due to the large and growing number of miRNA species and the huge number of putative targets. In silico analysis using prediction databases is the common method of identifying the potential target genes for miRNAs [19]. Therefore, further understanding of the multiple processes and pathways involved in pain processing [20] and their relationship to miRNA expression could have important clinical implications for CP management.

The purpose of this narrative review is to provide a comprehensive overview of human studies, examining the expression of miRNAs in the pathogenesis of CPP and CSP conditions. In addition, for those dysregulated miRNAs most consistently reported in the reviewed studies as being implicated in CP, bioinformatics analysis was then performed to investigate the dysregulated miRNA target genes and the signaling pathways involved, which may enhance our understanding of the potential pathophysiological impact of miRNAs in CP and of the molecular mechanisms leading to CP.

## 2. Results

In the following paragraphs, we have described some of the retrieved studies on miRNA expression distinguishing CPP and CSP. No data were found on miRNAs in chronic cancer-related pain, chronic secondary headache or orofacial pain. Table 2 and Table 3 summarize all studies, including information on miRNA extraction and detection methods, study design, sample type and tissue. The nomenclature of miRNAs was reported adopting the most recent official version used in miRbase [21].

### 2.1. miRNA Expression in Relation to Chronic Primary Pain

#### 2.1.1. Chronic Widespread Pain

Chronic widespread pain (CWP) is diffuse musculoskeletal pain in at least 4 of 5 body regions and in at least 3 or more body quadrants, as well as the axial skeleton [22]. Fibromyalgia (FM) is a form of CWP characterized by a reduced pain threshold and is regularly accompanied by fatigue, sleep disorders and depressive episodes [23]; it is a multifactorial disease whose pathophysiology is not well understood.

Braun et al. showed a downregulation of miR-103a-3p, miR-107 and miR-130a-3p and an upregulation of miR-125a-5p in FM patients compared to a control no-FM group. Moreover, FM patients were grouped according to the phenotypic characteristics of the disease, and only miR-103a-3p was expressed differently among clusters of FM patients. In particular, up-regulated miR-103a-3p and miR-107, which play a regulatory role in inflammation, were associated with adaptive coping in FM patients and the expression of miR-103a-3p correlated with FM-related disability [24].

In another study [25], six circulating miRNAs were significantly and differentially regulated in patients with FM compared to healthy controls. To identify putative gene targets of the deregulated miRNAs, a bioinformatic analysis was conducted, showing that several biological processes related to brain function/development and muscular functions were targeted by the dysregulated miRNAs in FM patients.

Leinders et al. [26] investigated systemic and cutaneous miRNA expression in FM patients. Several miRNAs were differently expressed compared to a control group, and only some of them were validated. Let-7d-5p, miR-103a-3p and miR-146a-5p resulted downregulated in the blood and skin of patients with reduced intraepidermal nerve fiber density (IENFD). Let-7d-5p expression in white blood cells showed positive correlations with IENFD and mean pain scores on the Graded Chronic Pain Scale in FM patients. In skin biopsies, let-7d-5p expression was higher in patients with FM, particularly in the subgroup with reduced IENFD. Furthermore, these patients showed lower expression of skin insulin-like growth factor-1 receptor (IGF-1Rs) than healthy controls. IGFs-1/IGF-1Rs are downstream targets of let-7d-5p (according to bioinformatic analysis) and play a major role in muscle regeneration [27].

#### 2.1.2. Complex Regional Pain Syndrome

Complex regional pain syndrome (CRPS) is characterized by allodynia, hyperalgesia, abnormal motor function and trophic disturbances [22].

A key feature of CRPS is neurogenic inflammation, and in fact in the study of Orlova et al., 14 differentially expressed miRNAs in patients with CRPS correlated with inflammatory and immune related markers, including vascular endothelial growth factor (VEGF), interleukin1 receptor antagonist (IL1Ra) and monocyte chemotactic protein-1 (MCP-1). VEGF and ILR1a, in particular, showed a significant correlation with the patients’ reported pain levels [28]. Douglas et al. showed that the miRNA profile was different between responders and poor responders both before and after ketamine therapy, an N-methyl-D-aspartate receptor agonist currently used for the treatment of CRPS. Among the differentially expressed miRNAs, miR-548d-5p, which was reduced more in poor responders than responders, showed two potential targets, namely UGT1A1 and CYP3A4, both important in drug metabolism. The functional analyses confirmed only UGT1A1, a member of the UDP-glucuronosyltransferase (GT) family. Moreover, the poor responders had a higher percentage of bilirubin than the responders, indicating that they could have increased levels of conjugated inactive glucuronides and therefore a greater removal of ketamine metabolites, potentially minimizing the therapeutic efficacy of ketamine [29]. Another dysregulated miRNA in ketamine-treated CRPS patients was miR-34a-5p, a miRNA with a known role in inflammation. In the study by Shenoda et al., an inverse correlation was observed between this miRNA and the long noncoding RNA X-inactive-specific transcript (XIST) in CRPS patients who responded poorly to therapy. Therefore, a decrease in miR-34a-5p could lead to an upregulation of XIST, which promotes inflammation via the pro-inflammatory transcription factor yin yang 1 [30].

Plasma exchange is an extracorporeal procedure used to purify the blood from molecules that can cause inflammation. In this regard, analysis of miRNAs in exosomes from the serum of responders and poor responders to plasma exchange showed that 17 miRNAs were significantly altered before and after plasma exchange. Among these miRNAs, miR-338-5p was down-regulated in poor responders compared to responders. The miR-338-5p target, i.e., IL-6 (a pro-inflammatory cytokine), was more reduced in responders following plasma exchange than before the therapy, thus directly contributing to the reduction of inflammation in CRPS. These results indicated that the combined approach of miRNA evaluation with cytokine analysis may be a feasible approach that can help predict the therapeutic efficacy of plasma exchange [31].

#### 2.1.3. Chronic Primary Headache or Orofacial Pain

A chronic primary headache or orofacial pain is defined as a headache or orofacial pain that occurs on at least 15 days per month for longer than 3 months and includes several subtypes [22].

Among them, trigeminal neuralgia (TN) is caused by peripheral nerve lesions and adversely affects the life of patients [32]. To date, only one study investigated the potential functions of miRNAs in human subjects with TN, showing an up-regulation of miR-132-3p, miR-146b-5p, miR-155-5p and miR-384 in patients with TN compared to healthy controls. Bioinformatic analysis revealed the potential involvement of these miRNAs in the onset and development of neuropathic pain. This result was confirmed by functional analysis indicating that miR-155-5p can directly target and downregulate nuclear factor-E2 related factor 2 (Nrf2), a factor that modulates the expression of genes with an implication in inflammation, immune response and antioxidant actions, all mechanisms involved in the development of TN [33].

Migraines, a disabling neurovascular disorder common in adults and children, is associated with activation and sensitization of the trigeminal pain circuit, vascular endothelial cells, inflammation and most likely dysregulation of glial cell activity [34].

Cheng et al. [35] conducted a study to evaluate the level of specific endothelial miRNAs in migraine patients. The miRNAs analyzed were chosen on the basis of their implication in the modulation of inflammation and oxidative stress in endothelial cells [36,37]. MiR-126-3p, miR-155-5p and let-7g-5p were found to be overexpressed in the migraine patients, while a similar but non-significant trend was observed for miR-21-5p. A correlation of miR-155-5p and miR-126-3p with syncope attack suggested that migraines could be a risk factor for syncope. In addition, the level of an endothelial dysfunction marker (intercellular adhesion molecule 1) showed a positive correlation with miR-155-5p, miR-21-5p and let-7g-5p in migraine patients. Altogether, these findings point out that migraine and vascular events may be associated with endothelial dysfunction [35].

In the study by Gallelli et al., overexpression of miR-34a-5p and miR-375 was found in young subjects with migraines and without aura who were not undergoing drug treatment, compared to the treated group and healthy subjects. The in-silico analysis showed that the target genes, i.e., *HCN3*, *NAV3*, *GPR158* for miR-34a-5p and *MTPN* for miR-375, are involved in trigeminal pain circuit of migraine [38].

Another problem in the evolution of migraines is the chronification that leads to increased disability. In this case, associated medication overuse could lead to reduced benefits. To study biomarkers involved in pathophysiology and the chronification of migraines, Greco et al. evaluated the expression of miR-34a-5p and miR-382-5p as well as that of calcitonin gene-related peptide (*CGRP*). The latter, due to its functions in cell degranulation, neurogenic inflammation and vasodilation, is involved in migraine headaches. The miRNAs analyzed were up-regulated in chronic migraine subjects with medication overuse compared to the episodic migraine group, and were positively correlated with *CGRP* levels. Moreover, the miRNA profile studied after detoxification from overused drugs showed reduced expression, in parallel with similar reduction also observed in *CGRP* levels. Hence, the medication overuse can modify the biology of migraine, including miRNA expression, and *CGRP* can play a role in migraine chronification [39].

#### 2.1.4. Chronic Primary Visceral Pain

Chronic primary visceral pain is a CPP localized in the head or neck, thoracic, abdominal or pelvic region, and is associated with related specific internal organs [22].

Irritable Bowel Syndrome (IBS) is a common chronic functional gastrointestinal disorder characterized by chronic abdominal pain and altered bowel habits, including diarrhea (IBS-D), constipation (IBS-C) or a mixture of both. IBS patients may suffer from visceral hypersensitivity and elevated visceral nociception [40], and several neurotransmitters are implicated in these processes, such as transient receptor potential vanilloid type 1 (TRPV1), serotonin reuptake transporter (SERT), etc. [41]. According to these premises, Zhou et al. found reduced expression of miR-199a-5p and miR-199b-5p in patients with IBS-D, which was related to the overexpression of TRPV1, a cation channel that, when activated, produces a burning sensation and/or pain [42]. Thus, the expression of these two miRNAs may contribute to chronic visceral pain and nociception through the up-regulation of TRPVI [41].

Another feature of IBS is the impaired intestinal mucosal barrier function, in which tight junctions and transmembrane proteins, including the cytosolic tight junction protein ZO-1 and claudin-1 (CLDN1), play an important role. In this regard, Zhu et al. investigated whether miR-29a-3p (a miRNA with potential implication in IBS) is associated with ZO-1 and CLDN1 and thus regulates intestinal mucosal barrier function. The level of miR-29a-3p was higher in patients with IBS-D than in healthy subjects, while the expression of ZO-1 and CLDN1 was lower, suggesting their potential role in the pathogenesis of IBS [43]. Another study on the association of intestinal barriers in IBS with miRNA expression was conducted by Mahurkar-Joshi et al., which found reduced expression of miR-219a-5p and miR-338-3p in IBS and IBS-C compared to healthy controls. The functional analysis showed that miR-219a-5p and miR-338-3p potentially contributed to altering barrier function and visceral hypersensitivity through dysregulated expression of proteasome/barrier function genes and of Mitogen Activated Protein Kinase (MAPK) signaling pathway genes, implicated in inflammation pain and hypersensitivity [44].

#### 2.1.5. Chronic Primary Musculoskeletal Pain

Chronic primary musculoskeletal pain (CPMP) is chronic pain experienced in muscles, bones, joints or tendons that cannot be directly attributed to a known disease or a damage process [22]. Among CPMP, multidisciplinary treatment programs are recommended as a first-line treatment for low back pain (LBP), given its multifactorial causes [45]. In this regard, the expression profiles of some miRNAs were evaluated in patients with chronic LBP and compared with those of healthy volunteers before the start of a multidisciplinary treatment program. Significant upregulation of miR-124-3p, miR-150-5p and miR-155-5p was found in patients with chronic LBP comparted to controls. Furthermore, the expression of these miRNAs was analyzed before and after treatment in responders and non-responders, and only the expression of miR-124-3p significantly increased in patients who responded to treatment [46]. MiR-124-3p regulates neurogenesis, neuronal differentiation and stress response [47], and is also considered one of the most important miRNAs for psychological disorders [48]. Therefore, according to the authors, the improvement in miR-124-3p expression may reflect the positive effect of multidisciplinary treatment on both pain and stress [46].

**Table 2 ijms-23-06016-t002:** MiRNA dysregulation related to chronic primary pain.

Type of Pain	miRNAs	miRNA Expression	Extraction Method	Detection Method	Study Design	Sample Size	Tissue	Reference
**Chronic widespread pain**								
*Fibromyalgia*	103a-3p, 107, 130a-3p 125a-5p	↓ ↑	miRNeasy Mini kit (Qiagen, Germany)	qRT-PCR	Cross-sectional	31 FM 16 HC	Blood	[24]
	21-5p, 23a-3p, 23b-3p 29a-3p, 99b-5p, 125b-5p, 145-5p, 195-5p, 223-3p	↓	miRNeasy Mini kit (Qiagen, Germany)	qRT-PCR	Cross-sectional	10 FM 8 HC	Cerebrospinal fluid	[49]
	let-7a-5p, 30b-5p, 103a-3p, 107, 151a-5p, 142-3p, 374b-5p 320a-3p	↓ ↑	miRNeasy Mini kit (Qiagen, Germany)	qRT-PCR	Cross-sectional	20 FM 20 HC	Serum	[50]
	143-3p, 145-5p, 223-3p, 338-3p, 451a	↓	mirVana miRNA isolation kit	qRT-PCR	Cross-sectional	11 FM 10 HC	PBMCs	[51]
	1-3p, 23a-3p, 133a, 139-5p, 320b, 346	↓	miRCURY RNA Isolation Kit-Biofluids	qRT-PCR	Cross-sectional	14 FM 14 HC	Serum	[25]
	let-7d-5p, 103a-3p, 146a-5p	↓	RNeasy Micro kit (Qiagen, Germany)	qRT-PCR	Cross-sectional	30 FM (12 reduced IENFD) 34 HC	Skin	[26]
**CRPS**	let-7a-5p, let-7b-5p, let-7c, 25-5p, 126-3p, 181a-2-3p,320a-3p, 320b, 532-3p, 625-5p, 629-5p, 664a-3p, 939-5p, 1285-3p	↓	PAXgene blood miRNA kit (Qiagen, Valencia, CA, USA)	TLDA	Cross-sectional	41 FM 20 HC	Whole blood	[28]
	33 miRNAs (before treatment) 43 miRNAs (after)	Dysregulated	PAXgene blood miRNA kit (Qiagen, Hilden, Germany)	TLDA	Cross-sectional	7 Responders Ketamine, 6 PR Ketamine	Whole blood	[29]
	34a-5p	↑	PAXgene blood RNA tubes (BD Biosciences, Franklin Lakes, NJ, USA)	qRT-PCR	Cross-sectional	7 Responders Ketamine, 6 PR Ketamine	Whole blood	[30]
	Let-7a-5p, 19b-3p, 19b-1-5p, 29b-3p, 30a-5p, 34b-5p, 126-5p, 191-5p, 195-5p, 338-5p, 484, 518b, 542-3p, 551b-5p, 618, 1183, 1274b	Dysregulated	miRvana miRNA isolation kit (Life technologies)	TLDA	Retrospective case series	3 RE-PE, 3 NR-PE	Serum	[31]
**Chronic primary headache or orofacial pain**								
*Trigeminal neuralgia*	132-3p, 146b-5p, 155-5p, 384	↑	TRIzol reagent (Invitrogen, Carlsbad, CA, USA)	qRT-PCR	Cross-sectional	28 TN 31 HC	Serum	[33]
*Migraine*	let-7g-5p, 126-3p, 155-5p	↑	miRNeasy Serum/ Plasma Kit (Qiagen, Germany)	qRT-PCR	Cross-sectional	30 MP 30 HC	Plasma	[35]
	27b-3p let-7b-5p, 22-3p, 181a-5p	↑ ↓	mirCURY LNATM Universal cDNA syn- thesis kit (Exiqon)	qRT-PCR	Cross-sectional	15 MP 13 HC	PBMC	[52]
	34a-5p, 375	↑	miRNeasy Serum/Plasma Kit (Qiagen, Venlo, The Netherlands)	qRT-PCR	Prospective parallel-group	12 untreated- MPWA 12 HC	Saliva	[38]
						12 untreated- MPWA 12 treated-MPWA	Saliva and serum	
	30a-5p	↑ in migraine	TRIzol reagent (TaKaRa, Otsu, Shiga, Japan)	qRT-PCR	Cross-sectional	Not reported	Serum	[53]
	34a-5p, 382-5p	↑	Direct-zol RNA Mini prep plus (Zymo Research from Aurogene, Rome, Italy)	qRT-PCR	Cross-sectional observational controlled	28 CM-MO 27 EM	PBMC	[39]
	29c-5p, 34a-5p, 382-5p	↑	RNeasy^®^ Plus 96 Kit (Qiagen, Hilden, Germany	qRT-PCR	Cohort	Cohort1: 8 MPA 8 MPPF	Serum	[54]
	29c-5p, 382-5p					Cohort1: 8 MPPF 8 HC		
	382-5p		miRNeasy^®^ Serum/Plasma Kit (Qiagen, Hilden, Germany)			Cohort2: 12 MPPF 12 HC		
**Chronic primary visceral pain**								
*Irritable bowel syndrome*	199a-5p, 199b-5p	↓	Trizol (Invitrogen, Carlsbad, CA, USA)	qRT-PCR	Cross-sectional	45 IBS-D 40 HC	Colon tissue	[41]
	24-3p	↑	miREasy Mini Kit Qiagen, Germany	qRT-PCR	Cross-sectional	10 IBS-D 10 HC	Intestinal mucosa epithelial cells	[55]
	29a-3p	↑	miRcute miRNA isolation kit (Tiangen Biotech Co., Ltd.)	qRT-PCR	Cross-sectional	21 IBS-D 16 HC	Sigmoid colon mucosa	[43]
	106b-5p, 532-5p 363-3p, 338-3p (IBS vs HC) 106b-5p 100-5p, 338-3p (IBS-C vs HC) 219-5p (IBS-D vs HC)	↑ ↓ ↑ ↓ ↓	Trizol (Invitrogen, Carlsbad, CA, USA)	nanoString nCounter Assay	Cross-sectional	29 IBS (15 IBS-C, 14 IBS-D) 15 HC	Colon tissue	[44]
	219-5p, 338-3p (IBS and IBS-C vs HC)	↓		qRT-PCR				
**Chronic primary musculoskeletal pain**	124-3p, 150-5p, 155-5p	↑	mirVana miRNA Isolation Kit (Ambion, Austin, TX, USA)	qRT-PCR	Prospective cohort	34 LBP 20 HC	CD4+ cells	[46]

Legend: CM-MO: chronic migraine-overuse of medications; EM: episodic migraine; HC: healthy controls; IBS-C: Irritable bowel syndrome-constipation; IBS-D: Irritable bowel syndrome- diarrhea; IC: interstitial cystitis; IENFD: intraepidermal nerve fiber density; LBP: low back pain; MP: Migraine patients; MPWA: Migraine patients without aura; MPA: migraine patients attack; MPPF: migraine patients pain-free; NR-PE: non-responders plasma exchange; PR: poor responder; qRT-PCR: Quantitative real-time PCR; RE-PE: responder plasma exchange; TN: trigeminal neuralgia. ↓/↑, difference in miRNAs expression, respectively up- and down-expressed.

### 2.2. miRNA Expression in Relation to Chronic Secondary Pain

#### 2.2.1. Chronic Post-Surgical Pain or Post-Traumatic Pain

One of the most common symptoms after major surgery is pain that, if undertreated, can lead to the development of persistent postoperative pain, increasing the risk of opioid addiction and related adverse effects [56]. The nociceptive and neuropathic components, often concomitant, are involved in the pathophysiology of postoperative pain. Human evidence specifically related to CSP to surgery is very scarce.

The exploratory study by Giordano et al. evaluated the potential use of circulating miRNAs as predictive biomarkers for chronic postoperative pain. The preoperative expression of circulating miRNAs, as validated or predicted to be associated with pain, inflammation and matrix degeneration, was measured in patients with severe knee osteoarthritis (OA) scheduled for total knee replacement and classified into two groups, the high-pain relief group and the low-pain relief group, based on the percentage of postoperative pain relief one year after surgery (more than 30% pain relief and less than 30% pain relief, respectively). Three miRNAs, miR-146a-5p, miR-145-5p and miR-130b-3p, were significantly dysregulated in the low-pain relief group compared to the high-pain relief group, and miR-146a-5p tended to be an independent predictor of postoperative pain relief [57]. Therefore, postoperative pain appeared to be associated with a specific preoperative serum miRNA signature.

The involvement of the inflammatory/oxidative stress modulatory miR-21-5p in post-surgical pain has been proposed in the context of surgical removal of the impacted mandibular third molar [58], where increased bone expression of miR-21-5p has been associated with a delayed postoperative pain onset, potentially reflecting a reduction in surgical (and inflammatory) stress. However, this study addressed short-term surgical outcomes including pain, and the role for miR-21-5p in chronic post-surgical pain needs to be determined.

Prospective studies have indicated that miRNA profiles can serve as molecular and blood-based predictors of chronic post-traumatic musculoskeletal pain. Indeed, 32 miRNAs from blood collected following a traumatic/stressful event such as a motor vehicle collision were significantly different among individuals who did and did not develop severe persistent musculoskeletal pain in the axial region (neck, shoulders, and/or back) 6 weeks after the trauma [59]. In particular, two miRNAs, miR-135a-5p (regulating serotonin receptor and transporter) and miR-3613-3p (regulating pain-associated genes), strongly predicted the subsequent development of CP. Most of these differentially expressed miRNAs are associated with stress response and/or pain processing, thus supporting their role in the pathogenesis of CP. Interestingly, the differentially expressed miRNAs were enriched for gene location on the X chromosome and appeared to contribute to persistent severe axial pain in women but not in men [59]. The same authors demonstrated that reduced levels of circulating miR-320a-3p in the peritraumatic period predicted persistent diffuse and axial musculoskeletal pain 6 months after a motor vehicle collision in subjects suffering from significant peritraumatic distress, possibly targeting stress and pain-associated transcripts, as demonstrated by bioinformatic analysis and in vitro luciferase reporter assays [60]. Furthermore, peritraumatic circulating levels of miR-19b-3p, which target pain and circadian rhythm related genes, also predicted widespread posttraumatic pain and posttraumatic stress symptoms at follow-up in a sex-dependent manner [61]. Indeed, an inverse relationship between miR-19b-3p expression levels and the probability of developing posttraumatic widespread pain and stress symptoms was shown in women, and a positive relationship between miR-19b-3p expression and the probability of developing these outcomes was shown in men.

Another prospective study was conducted on the whole blood of individuals who reported neck pain related to motor vehicle collisions. Biologic pathways targeted by pain-associated miRNAs include extracellular matrix receptor interaction, TGF-β signaling, amphetamine addiction and axon guidance, all contributing to stress, pain pathogenesis and persistent posttraumatic disability related to neck pain. Furthermore, let-7i-5p expression, previously associated with adipose tissue function, appeared to mediate the association between neck muscles fat infiltration and neck-pain related disability [62].

#### 2.2.2. Chronic Neuropathic Pain

Chronic neuropathic pain (CNP), as previously mentioned, is typically perceived within the innervation territory, which is somatotopically represented within the structure of the damaged nervous system [3].

Several studies have examined the signature of miRNAs in CNP. Liu et al. [63] reported the downregulation of miR-101-3p in both plasma and sural nerves of subjects with neuropathic pain of various origins. Suppression of miR-101-3p has been correlated with pain pathophysiology through inhibition of miR-101-3p-mediated nuclear factor kappa B (NF-κB) activation, a key contributor to the neuroinflammation involved in the pathogenesis of CNP [64]. In accordance with the role of inflammation in the pathogenesis of CP in peripheral neuropathies, inflammation-associated miRNAs were found to be specifically deregulated in peripheral neuropathies of different etiology in comparison with healthy controls, with miR-21-5p (a key regulator of immunological processes) up-regulated in white blood cells and sural nerve especially in painful neuropathies, miR-146a-5p (with anti-inflammatory action) up-regulated in white blood cells and down-regulated in distal skin especially in painful neuropathies and miR-155-5p (otherwise known as pro-inflammatory) down-regulated in white blood cells and distal skin especially in painful inflammatory neuropathies and up-regulated in the sural nerve of painful neuropathies. Strikingly, the aberrant nerve miRNA expression correlated with pain in peripheral neuropathies [65]. Taken together, these results suggest a complex reciprocal neuro-immune interaction at both the systemic and local levels, in which miRNAs may play an active modulating role.

Patients in the acute phase of Herpes Zoster infection compared to controls showed 6 significantly up-regulated miRNAs (miR-190b, -571, -1276, -1303, -661, -943) in serum samples, which were related to the regulation of immunity and inflammation as well as nervous system development. About one-tenth of patients with acute Herpes Zoster develop post-herpetic neuralgia, which is attributed to sensory nerve damage and may persist for at least 3 months [66].

Intervertebral disc degeneration (IDD) is a common degenerative disease whose main manifestations are lower back pain and lumbar disc herniation (LDH), which weigh heavily on the global health system. The pathogenesis of IDD involves a pro-inflammatory reaction with recruitment and activation of immune cells, release of inflammatory factors including IL-6 and TNF-α, extracellular matrix degradation, cell apoptosis, altered autophagy, vascular proliferation and nerve cell degeneration, which leads to nerve root sensitization and CNP [67].

In the study by Cui et al., the signature of circulating miRNAs of patients with IDD and manifest LDH was analyzed. A total of 73 dysregulated miRNAs resulted compared to a control group; among them, upregulated miR-766-3p and miR-6749-3p and downregulated miR-4632-5p can target multiple genes related to IDD and LDH pathogenesis. In particular, bioinformatic analysis identified many pathways, such as endocytosis, apoptosis, axon guidance, vascular endothelial growth factor (VEGF), cell adhesion molecular, focal adhesion, extracellular matrix (ECM)-receptor interaction, phosphoinositide 3-kinase (PI3K)/Akt, mTOR and NF-κB pathways, showing the involvement of differentially expressed miRNAs in the modulation of inflammation, proliferation, differentiation and oxidative stress [68].

The pathogenesis of IDD was studied by Dong et al. [69], who collected nucleus pulposus (NP) and annulus fibrosus cells (which are intervertebral disc components) from patients with IDD; patients with idiopathic scoliosis were used as a control. MiR-640 was significantly upregulated only in NP cells from patients with IDD. In vitro experiments showed that inflammatory cytokines (TNF-α and IL-1β) were also elevated in NP cells, stimulating miR-640 expression via the NF-κB signaling pathway. This may generate a positive feedback loop to aggravate inflammation in IDD, as a high level of miR-640 enhanced TNF-α and IL-1β production. MiR-640 also repressed the Wnt signaling pathway, which has important functions during senescence and apoptosis, suggesting that this miRNA is implicated in the degeneration of NP cells and could be used as a potential therapeutic target for the biotherapy of low back pain.

Given the neuropathic and neuroinflammatory nature of disc herniation-related leg pain, Hasvik et al. investigated the expression of miR-17-5p belonging to the miR-17-92 cluster and influencing the development of neuropathic pain conditions. A total of 97 patients were followed up for 1 year after disc herniation, and elevated serum miR-17-5p expression was shown to be associated with increased leg pain intensity. After an in vivo experiment using a rat model, it has been suggested that the high expression of miR-17-5p in the CNP facilitates the local release of TNF from circulating immune cells via a pro-inflammatory mechanism, which increases pain intensity at the legs after disc herniation [70].

#### 2.2.3. Chronic Secondary Visceral Pain

Chronic secondary visceral pain is a persistent or recurrent pain that originates from internal organs of the head/neck region and the thoracic, abdominal and pelvic cavities. It includes bladder pain syndrome (BPS), chronic prostatitis, endometriosis, etc. [3].

BPS is a clinical syndrome of pelvic pain and urinary frequency and urgency without infection. A possible cause of BPS can be epithelium dysfunction resulting in leakage of the bladder urothelium. Several molecules are involved in the regulation of epithelial permeability and bladder contractility and inflammation, and their downregulation may indicate impaired tight junction structure and increased permeability of the bladder urothelium [71]. Among these down-regulated molecules, Sanchez Freire et al. [72], in their study, showed a significantly lower expression of neurokinin 1 (NK1) tachykinin receptor in BPS patients than in controls. Moreover, miR-499b-5p, miR-500a-5p, miR-328-3p and miR-320a-3p were upregulated in biopsies of BPS patients and negatively correlated with NK1R mRNA and/or protein levels, indicating that BPS can lead to an attenuation of NK1 tachykinin receptor synthesis via activation of these miRNAs [72]. In the study by Arai et al. [73], small RNA sequencing performed on interstitial cystitis tissues, normal bladder tissue and bladder cancer revealed the down-regulation of miR-320 family in interstitial cystitis tissues. The in silico analysis carried-out on this miR-family showed 85 putative targets, but the authors focused on three transcription factors, namely E2F transcription factor 1, E2F transcription factor 2 and tubby bipartite transcription factor, due to their potential to significantly influence downstream RNA networks. The immunohistochemistry assay on these proteins displayed greater expression in interstitial cystitis cells than in normal bladder and bladder cancer tissues. Considering the functions of these three proteins, their overexpression could lead to the inflammation observed in bladder pain syndrome [73].

Prostatitis is another visceral pain syndrome made up of 4 main categories: non-bacterial acute prostatitis, bacterial chronic prostatitis, chronic prostatitis/chronic pelvic pain syndrome (CP/CPPS) and asymptomatic inflammatory prostatitis [74]. The study by Chen et al. demonstrated the presence of deregulated miRNAs in the prostatic secretion of CP/CPPS patients, and miR-21-5p, miR-103a-3p and miR-141-3p were further validated, demonstrating that miR-21-5p possessed a high classify-accuracy for CP/CPPS patients with a significant pain score [75].

Regarding endometriosis, a gynecological disease characterized by the presence of endometrial-like tissue outside the uterus that induces inflammatory reactions, in the study by Liang et al. [76], miR-200c-3p was found to be downregulated in ectopic endometrial tissues compared to normal endometrial tissue. Furthermore, the expression of metastasis-associated lung adenocarcinoma transcript 1 (MALAT1) was upregulated in ectopic endometrial tissues compared to normal endometrial tissues, and the level of MALAT1 was negatively correlated with the level of miR-200c-3p. Thus, miR-200c-3p may suppress the proliferation and migration of endometrial stromal cells by downregulating MALAT1, which in turn functions as a competing endogenous RNA to upregulate the expression of *Zinc Finger E-Box Binding Homeobox 1* and *2* by sequestering miR-200c-3p. The MALAT1/miR-200c sponge could act as a novel target for the diagnosis and treatment of endometriosis [76].

Razi et al. compared the plasma level of miR-185-5p in women with endometriosis and a control group. This study showed that there was significantly lower expression of miR-185-5p in the plasma of the case group compared to controls, but no significant difference in the expression of VEGF and PDGF (platelet-derived growth factor) mRNAs, two proangiogenic factors involved in endometriosis and predicted targets of miR-185-5p. Therefore, circulating VEGF and PDGF levels appear to be unrelated to endometriosis, while miR-185-5p could potentially serve as a candidate biomarker for endometriosis [77].

#### 2.2.4. Chronic Secondary Musculoskeletal Pain

Chronic secondary musculoskeletal pain is a chronic musculoskeletal pain that arises from an underlying disease [78]. The study by Dayer et al. [79] showed that some miRNAs were expressed differently depending on the origin of the pain. 10 miRNAs were validated in patients diagnosed with CP of either neuropathic or nociceptive origin. Among these, 7 miRNAs showed significant differential expression between the nociceptive and neuropathic groups. Moreover, miR-320a and miR-98-5p were able to discriminate between the two groups of CP, thus suggesting the potential of these miRNAs as diagnostic tools. Regarding the biological significance of these miRNA changes in CP, some speculations have been made by the authors, including the role in inflammation for miR-98 and miR-205, and specifically in neuropathic pain for let-7d, miR-29c and miR-320, as previously demonstrated [79].

Osteoarthritis (OA), the most common form of arthritis, is a chronic painful disease of the synovial joints associated with cartilage destruction and inflammation leading to motor disability [80]. The mechanism’s underlying pain in OA are multifactorial, including nociceptive, inflammatory and neuropathic components, and are patient- and time-specific, thus complicating the treatment strategy [81]. Angiogenesis, the formation of new blood vessels from pre-existing ones, can promote synovial inflammation and pain, and, in fact, angiogenesis inhibitors have been shown to reduce pain in OA models [82]. According to this premise, Xie et al. [83] found that the angiogenesis-related miR-210, known to regulate the expression of the pro-angiogenic factor VEGF, was upregulated in synovial fluid samples from patients with both early- and late-stage OA compared to healthy subjects, in correlation with the increased protein expression of VEGF. Although further validation is required, the early change in miR-210 expression in OA suggests its potential as an early diagnostic biomarker for OA.

In another study, the expression of 18 miRNAs potentially associated with OA pathology in serum and cartilage samples obtained from patients with OA was examined [84]. Among the tested miRNAs, miR-146a-5p expression was significantly upregulated in both tested OA patients’ samples compared to healthy controls, with a positive correlation between levels in both sample types. This finding suggests that miR-146a-5p serum levels could reflect molecular processes in cartilage, thus potentially representing a biomarker for OA [84]. These results are in line with previous data documenting an increase in the expression level of miR-146a-5p in the cartilage of patients with OA, its induction under inflammation and the reported role for this miRNA in OA pathogenesis and pain through the regulation of genes involved in synovial inflammation, neoangiogenesis and osteoclastogenesis.

MiR-34a-5p, another miRNA associated with cellular processes linked to the pathogenesis of OA, such as inflammation, cell growth and apoptosis, was found to be up-regulated in cartilage samples as well as inflamed primary chondrocytes from patients with OA compared to controls [85]. Its induction has been associated with inflammation-related down-regulation of the long non-coding RNA SNHG7 (small nucleolar RNA host gene 7), which targets miR-34a-5p. As a consequence of the upregulation of miR-34a-5p, the expression of its target synoviolin 1 (SYVN1), which is involved in endoplasmic reticulum stress, chronic inflammation and vascular overgrowth, was reduced [85]. Although the precise role of miR-34a-5p in pain development and/or maintenance is unknown, it shows several pain-relevant targets, including *SCN2B*, *KCNK3*, *CACNA1E* and soluble N-ethylmaleimide-sensitive factor (NSF) attachment protein receptors (SNAREs) such as VAMP-2, syntaxins and SNAP-25, some of which have been shown, at least in animal models, to be potentially involved in neuropathic [86] and inflammatory pain [87].

Recently, a molecular mechanism in OA has been proposed from clinical and in vitro evidence in which extracellular matrix degradation and chondrocytes apoptosis have been associated with upregulation of the transcription factor lymphoid enhancer-binding factor 1 and the subsequent induction of the circular RNA circRNF121, a non-coding regulatory RNA that targets miR-665, thereby releasing its inhibition on the MYD88/NF-κB pro-inflammatory pathway. Concordantly, reduced levels of miR-665 were found in the cartilage tissues of patients with OA compared to normal tissues. Although the role of miR-665 in OA-related inflammation and degenerative processes has been postulated in this study [88], its function in OA-associated pain processes remains unclear.

A recent prospective study looked for potential prognostic miRNAs based on the change in miRNA expressions after elevated tibial osteotomy in the medial OA compartment of the knee. Two miRNAs, miR-23a-3p and miR-30c-5p, were significantly altered after high tibial osteotomy compared to the preoperative period. Changes in clinical symptoms and radiological measurements were then used to identify specific miRNAs related to better outcomes. Only miR-30c-5p showed significant correlation with the postoperative pain relief [89]. Although the clinical implications and related mechanisms of these changes in miRNAs expression remain to be conclusively demonstrated, the results indicate the potential of these miRNAs as biomarkers for clinical improvement after a high tibial osteotomy.

**Table 3 ijms-23-06016-t003:** MiRNA dysregulation related to chronic secondary pain.

Type of Pain	miRNA	miRNA Expression	Extraction Method	Detection Method	Study Design	Sample Size	Tissue	Reference
**Post-surgical pain**	130b-3p, 145-5p, 146a-5p	↓ (low-pain relief group vs high-pain relief group)	miRNeasy Serum/Plasma Advanced Kit (QIAGEN, Hilden, Germany)	qRT-PCR	Prospective (1 year)	136 patients with a knee OA scheduled for total knee replacement, divided at follow-up in 22 low-pain relief group, and 114 high-pain relief group	Serum	[57]
**Post-traumatic pain**	135a-5p, 3613-3p, 19b-3p, 502-3p 7-5p, 26b-3p, 185-5p	↑ ↓	PAXgene blood miRNA kit (Qiagen, Valencia, CA, USA)	qRT-PCR	Prospective (6 weeks)	53 subjects with a motor vehicle collision, who developed (27) or not (26) persistent axial pain at follow-up	Whole blood	[59]
	320a-3p	↓ (according to presence/severity of persistent axial and widespread musculoskeletal pain)	PAXgene blood miRNA kit (Qiagen)	Sequencing and qRT-PCR	Prospective (6 weeks)	69 subjects with a motor vehicle collision, assessed for presence/severity of different axial and widespread musculoskeletal pain at follow-up	Whole blood	[60]
	19b-3p	↓ (women) ↑ (men)	PAXgene blood miRNA kit (Qiagen, Germantown, MD, USA)	Sequencing and qRT-PCR	Prospective (6 weeks)	179 subjects with a motor vehicle collision, assessed for persistent posttraumatic widespread pain at follow-up	Whole blood	[61]
	14 miRs	Deregulated	PAXgene blood microRNA kit (Qiagen, Germantown, MD, USA)	Sequencing and qRT-PCR	Prospective (1 year)	43 subjects with a motor vehicle collision, with different neck pain outcomes during follow-up	Whole blood	[62]
**Chronic Neuropathic pain**	101-3p	↓	Trizol Reagent (Invitrogen, Carlsbad, CA, USA)	qRT-PCR	Cross-sectional	32 CNP 10 HC	Plasma and sural nerve biopsies	[63]
	132-3p	↑	miRNEASY kit (Qiagen, Hilden, Germany)	qRT-PCR	Cross-sectional	30 patients with peripheral neuropathy, 81 patients with painful or painless inflammatory or non-inflammatory neuropathies 30 HC	White blood cells and sural nerve biopsies	[90]
	21-5p, 146a-5p, 155-5p	Deregulated	Not reported	qRT-PCR	Cross-sectional	76 patients with peripheral neuropathies (of which: 24 with inflammatory neuropathy, 31 with non-inflammatory neuropathy, 21with neuropathy of unknown etiology, 39 with a painful neuropathy and 37 with painless neuropathy) 30 HC	White blood cells, sural nerve, and skin punch biopsies	[65]
	190b, 571, 1276, 1303, 661, 943	↑	TRIzol reagent (Invitrogen, San Diego, CA, USA)	TLDA and qRT-PCR	Cross-sectional	41 acute Herpes Zoster patients, 35 HC	Serum	[66]
	34c-5p, 107, 127–5p, 486–3p, 892b	↑	Trizol Reagent (Invitrogen, Carlsbad, CA, USA)	TLDA and qRT-PCR	Cross-sectional	37 patients with acute Herpes Zoster, 29 patients with post-herpetic neuralgia	Serum	[91]
	223-3p	↑	miRNeasy isolation kit (Qiagen)	qRT-PCR	Prospective (1 year)	97 patients with lumbar radicular pain after disc herniation	Serum	[92]
	21-5p	↑	Trizol Reagent (Invitrogen, Carlsbad, CA, USA)	qRT-PCR	Cross-sectional	10 patients with lumbar disc herniation accompanied by nerve root pain, 10 patients with lumbar burst fractures (HC)	Nucleus pulposus tissues of intervertebral disc herniation	[93]
*Intervertebral disc degeneration*	17 miRNAs 56 miRNAs	↑ ↓	TRIzol^®^ reagent (Invitrogen, Carlsbad, CA, USA)	Sequencing	Cross-sectional	10 IDD 10 HC	Serum	[68]
	130b-3p, 200c-3p 10a-5p, 25-3p, 34a-5p, 182-5p	↑ ↓	miRNeasy mini kit (Qiagen, Valencia, CA, USA)	qRT-PCR	Cross-sectional	3 IDD 3 HC	Disc	[94]
	640	↑	TRIzol (Invitrogen, 15596018)	qRT-PCR	Cross-sectional	15 IDD 5 HC	NP and AF cells	[69]
	185-5p	↓	miRNeasy Mini kit (Qiagen GmbH)	qRT-PCR	Cross-sectional	10 IDD 10 HC	NP cells	[95]
	17-5p	↑	miRNeasy serum plasma isolation kit (Qiagen, Hilden, Germany)	qRT-PCR	Cohort	97 patients with lumbar radicular leg pain and disc herniation	Serum	[70]
**Chronic secondary visceral pain**								
*Bladder syndrome*	320-a-3p, 328-3p, 499b-5p, 500a-5p	↑	mirVana miRNA isolation kit (Ambion)	TLDA	Cross-sectional	8 BPS 4 HC	Dome bladder biopsies	[72]
	320a-3p, 320b, 320c, 320d	↓	TRIzol reagent (Invitrogen, Carlsbad, CA, USA)	Sequencing	Cross-sectional	8 IC cases, 5 normal bladder, 5 bladder cases	IC tissues	[73]
*Prostatitis*	21-5p, 103a-3p, 141-3p	↑	miRCURY RNA isolation kit (Exiqon, Woburn, MA, USA)	qRT-PCR	Prospective study	21 IIIA CP/CPPS patients	Prostatic secretion	[75]
*Endometriosis*	185-5p	↓	QIAamp miRNA Plasma/Serum Mini kit (QIAGEN, Germany)	qRT-PCR	Case-control	25 patients 25 HC	Plasma	[77]
	let-7a-5p, let-7b-5p, let-7d-5p, let-7f-5p, let-7g-5p, let-7i-5p, 199a-3p, 320a-3p, 320b, 320c, 320d, 320e, 328-3p, 331-3p	↓	Trizol reagent	qRT-PCR	Cross sectional	19 OE patients 21 HC	Plasma	[96]
	200c-3p	↓	TRIzol reagent (Invitrogen, Carlsbad, CA, USA)	qRT-PCR	Cross-sectional	27 endometriosis patients, 12 uterine leiomyoma or hysterectomy in patients with grade II–III cervical intraepithelial neoplasia	Ectopic endometrial tissues	[76]
	9-5p, 34b-3p, 34c-5p	↓	Trizol reagent (Invitrogen, Carlsbad, CA, USA)	qRT-PCR	Cross-sectional	4 patients 3 HC	Endometrial biopsies	[97]
**Chronic secondary musculoskeletal pain**	let-7d-5p, 98-5p 29c-3p, 205-5p, 222-3p, 320a-3p, 423-5p	↑ (No vs Np) ↓ (No vs Np)	Exiqon miRCURY RNA isolation (Exiqon, Vedbaek, Denmark)	qRT-PCR	Cross-sectional	100 patients (60 No, 40 Np)	Plasma	[79]
	210	↑	Isothiocyanatephenol/chloroform extraction	qRT-PCR	Cross-sectional	20 early-stage OA patients, 20 late-stage OA patients 10 HC	Synovial fluid from knee joints	[83]
	146a-5p	↑	For serum: miRCURY RNA Isolation Kit–Biofluids (Exiqon, Denmark); for cartilage: miRVana miRNA isolation kit (Applied Biosysstems, Life Technologies, Carlsbad, CA, USA)	qRT-PCR	Cross-sectional	28 OA patients 2 HC	Cartilage tissue and serum	[84]
	34a-5p	↑	TRIzol reagent (Invitrogen, Carlsbad, CA, USA)	qRT-PCR	Cross-sectional	15 OA patients 10 HC	Cartilage tissue and cultured primary chondrocytes	[85]
	665	↓	RNeasy Mini Kit (Qiagen, Valencia, CA, USA)	qRT-PCR	Cross-sectional	30 OA patients 5 HC	Cartilage tissue and cultured primary chondrocytes	[88]
	let-7e-5p, 454-5p 885-5p	↓ ↑	miRNeasy kit (Qiagen, Inc., Valencia, CA, USA)	qRT-PCR	Prospective cohort (Bruneck study)	entire cohort (816 subjects)	Serum	[98]
	16-5p, 29c-3p, 93-5p, 126-3p, 146a-5p, 184, 186-5p, 195-5p, 345-5p, 885-5p	↑	miRNeasy kit (Qiagen, Inc., Valencia, CA, USA)	qRT-PCR	Cross-sectional	27 OA patients 27 HC	Plasma	[99]
	9-5p, 138-5p, 146a-5p, 335-5p	↑	mirVana miRNA Isolation Kit (Exiqon, Vedbaek, Denmark)	qRT-PCR	Cross-sectional	40 early and late OA patients 2 HC	Cartilage	[100]
	146a-5p, 155-5p, 181a-5p, 223-3p	↑	Trizol reagent (Invitrogen, Carlsbad, CA, USA)	qRT-PCR	Cross-sectional	36 OA patients 36 HC	PBMC	[100]
	23a-3p, 30c-5p	↑	miRNeasy Serum/Plasma Kit (Qiagen, Germany)	qRT-PCR	Prospective (before and 6 months after high tibial osteotomy)	22 patients with medial compartmental knee OA	Synovial fluid	[89]

Legend: BPS: bladder pain syndrome; CNP: chronic neuropathic pain; HC: healthy controls; IC: Interstitial cystitis; IDD: Intervertebral disc degeneration; No: nociceptive pain; Np: neuropathic pain; OE: ovarian endometriosis; OA: osteoarthritis; NP: nucleus pulposus; PBMC: peripheral blood mononuclear cell; qRT-PCR: Quantitative real-time PCR; TLDA: TaqMan Low Density Array. ↓/↑, difference in miRNAs expression, respectively up- and down-expressed.

### 2.3. Bioinformatic Analysis of miRNA Gene Targets and Pathways

Using the miRNAs found in at least 3 studies, i.e., for CPP: let-7a-5p, hsa-miR-34a-5p, hsa-miR-103a-3p, hsa-miR-155-5p; for CSP: hsa-miR-146a-5p, hsa-miR-185-5p, hsa-miR-320a-3p; we obtained the following targets: let-7a-5p: 632 targets; hsa-miR-34a-5p: 419 targets; hsa-miR-103a-3p: 388 targets; hsa-miR-155-5p: 145 targets; 146a-5p: 58 targets; hsa-miR-185-5p: 165 targets; hsa-miR-320a-3p: 377 targets. As created in Cytoscape, the network consists of 1866 nodes and 2183 edges (Figure 1). Detailed information about the miRNA-gene target is listed in Appendix A). Considering that mRNAs are typically targeted by many miRNAs and that each miRNA targets multiple mRNAs, the miRNA-gene pairs identified are part of co-regulation and interaction networks where different mRNA targets overlap between different miRNAs and between CPP and CSP.

Then, we performed a KEGG pathway analysis to identify the possible signaling pathways related to CPP and CSP. The analysis was performed by separating the two types of pain in order to potentially discriminate between CPP and CSP. Therefore, for a more robust and convincing analysis, targets of all miRNAs were taken together, and duplicates were removed. In that way, 1438 overlapping targets for CPP and 566 overlapping targets for CSP are identified and analyzed with GeneTrail2.

We identified in total 131 and 77 KEGG pathways for CPP and CSP, respectively. A deeper insight into these pathways showed that the vast majority is associated with cellular processes. In particular, for CPP we found pathways involved in signal transduction such as PI3K-Akt signaling, MAPK signaling pathway, RAP1 signaling pathway and Ras signaling pathway; in the immune system, such as Fc gamma R-mediated phagocytosis, Leukocyte transendothelial migration and a Chemokine signaling pathway and in the endocrine system, such as melanogenesis and the insulin signaling pathway (Figure 2). Additionally, for CSP we found pathways involved in signal transduction, such as the ErbB signaling pathway, the mTor signaling pathway, MAPK signaling pathways, PI3K-Akt signaling pathways and the Rap1 signaling pathway; in the immune system, such as the Toll-like receptor signaling pathway, the Chemokine signaling pathway and in the endocrine system, such as the Insulin signaling pathway (Figure 3). Interestingly, some of the main pathways for CPP are also related to the function of the nervous system, such as the Neurotrophin signaling pathway, Dopaminergic synapse and Cholinergic synapse, and to substance dependence, such as amphetamine addiction, morphine addiction and cocaine addiction (Figure 2). For both CPP and CSP some other pathways are associated with diseases such as pathways in cancer and viral infectious diseases (Figure 4 and Figure 5). See Appendix A for further details (Appendix A).

## 3. Discussion

To the best of our knowledge, this is the first review to summarize the human studies on miRNA expression associated with CPP and CSP conditions, and the first to highlight the potential biological mechanisms leading to pathological conditions. After peripheral inflammation or nerve injury, the increase in inflammatory mediators causes a change in the expression of some miRNAs, resulting in an alteration in pain-related genes. Such an alteration leads to an increase in dorsal root ganglion neuronal excitability, spinal central sensitization and hyperalgesia and allodynia, hence the chronic pain.

The data presented here indicate that the dysregulation of some miRNAs is associated with the pain condition, suggesting their potential role in the pathogenesis of certain diseases and the pain process. For example, miR-34a-5p, already known for its role in regulating the inflammatory response and vascular endothelial stress response [101], could be considered as a potential candidate in the pathogenesis of migraines [38,39,54]. It is involved in pathways regulating the function of nervous system, such as the Dopaminergic synapse and GABAergic signaling. Andersen et al. found that this miRNA negatively modulated the expression of genes that facilitate GABAergic signaling, suggesting a facilitator transmission that promotes the nociceptive effect [54]. Moreover, miR-34a-5p could be considered as biomarker for therapeutic response prediction. It is known that drug administration can indirectly affect miRNA expression [102], and some specific miRNAs could be relevant as indicators of a drug response [103]. A decreased expression of miR-34a-5p was found in the study of Gallelli et al., in patients under drug treatment compared to untreated subjects [38]. A growing body of evidence indicates that miRNAs can be useful in monitoring therapeutic responses, and knowledge of the involvement of miRNAs in the induction and progression of CP may open the opportunity to develop miRNA-based therapeutics as a new analgesic modality. Of course, there are still many challenges, including the rational selection of miRNAs to target, the delivery system and the route of administration (systemic vs tissue), as well as the stage of painful disease in which targeted miRNAs play a role. Another prominent miRNA featured in this review was miR-103a-3p, belonging to the so-called miR-15/107 group of miRNA genes involved in the regulation of metabolism, cell division, stress responses and exercise [104]. This miRNA was mainly found in studies conducted in patients with FM [24,26,50] and is involved in different pathways such as the SNRK/NF-κB/p65 signaling pathway [105]. An elevated gene expression of miR-103a-3p can lead to the suppression of SNRK and to an over-activation of the pro-inflammatory transcription factor NF-κB. This machinery is linked to mechanisms promoting epigenetic resilience and strategies to limit damage due to diseases [106], as well as synaptic plasticity, which results in adaptive processes [107]. MiR-155-5p has here emerged as being associated with CPP, in particular with TN, migraines and musculoskeletal pain. Its expression is induced by factors associated with infections, injury, hypoxia and inflammation, through signaling pathways involving NF-κB activation; contrarily, it is decreased by anti-inflammatory cytokines, resolvins and glucocorticoids, thus providing a feedback inhibition of the miR-155-induced responses [108]. MiR-155-5p plays a multifunctional role in innate and adaptive immunity and inflammation, so that its aberrant overexpression represents a potential disease biomarker or therapeutic target in states of chronic inflammation [109]. Its involvement in neuroinflammation, vascular dysfunction and neurodegeneration has been documented in animal and human studies [110,111]. Interestingly, a miR-155-5p target, i.e., NRG3 (neuregulin-3), is a ligand of ERBB4, and is thought to regulate neuroblast proliferation, migration and differentiation; repair after nerve injury; synapse formation; regulation of glutamate; GABA and dopamine release [112]. Recent studies point to an important role for miR-155-5p in the development of neuropathic pain and associated mechanical allodynia, thermal hyperalgesia and neuronal inflammation, through mechanisms involving the regulation of the suppressor of cytokine signalling 1 (SOCS1) and TNF-α receptor-transient receptor potential ankyrin 1 (TNFR1-TRPA1) pathways [113,114,115].

Another miRNA related to CPP is let-7a-5p, a highly evolutionarily conserved miRNA that resulted in downregulation in patients with FM and CRPS. Let-7a-5p has been reported to regulate stem-cell differentiation, neuromuscular development and cell proliferation and differentiation, and, like other members of the let-7 family, it seems to function as a tumor suppressor [116,117]. The upregulation of let-7a-5p in microglia has been found to correlate with protection from ischemia and neuroinflammation by inhibiting the expression of pro-inflammatory factors (e.g., IL-6, iNOS) and boosting the expression of anti-inflammatory factors (IL-10 and IL-4) [118]. Interestingly, in cellular and animal models, chronic treatment with morphine led to an increase in brain let-7 miRNAs expression and to a parallel decrease in the expression of μ-opioid receptor, thus implicating let-7 in the mechanism of opioid tolerance [119], which limits the analgesic efficacy of opioids.

A downregulation of miR-185-5p was observed in CSP conditions including IDD, chronic secondary visceral pain and post-traumatic pain. MiR-185-5p has been found to protect against excessive extracellular matrix degradation associated with inflammation and degeneration in human IDD samples: indeed, here, the higher expression of a circular RNA derived from tissue inhibitor of metallopeptidases 2 (circ-TIMP2) led to the sequestration of miR-185-5p and the consequent upregulation of miR-185-5p target genes, including *MMP-2*, associated with extracellular matrix catabolism [95]. These data are in agreement with other animal findings in which miR-185-5p suppressed autophagy and apoptosis of nucleus pulposus cells in IDD via inhibition of the Wnt/β-catenin signaling pathway [120]. Furthermore, anti-inflammatory effects have been observed for miR-185-5p in neuropathic pain induced by experimental chronic constrictive injury, where miR-185-5p was reduced in spinal cord tissues and its upregulation attenuated mechanical and thermal hyperalgesia and decreased microglia and astrocytes recruitment and the levels of inflammatory cytokines by repressing NLRP3 inflammasome [121].

Another pain-associated miRNA is miR-320a-3p, which has here emerged as mostly associated with CSP; an important finding from the study by Dayer et al. was its predictive value, in combination with miR-98-5p, for the discrimination between the nociceptive and neuropathic origin of pain [79], so that it resulted in downregulation and upregulation, respectively. Similarly, its expression is deregulated in secondary visceral pain conditions, as well as in post-traumatic pain. Downregulation of miR-320a-3p might lead to heightened inflammation through the upregulation of its target’s MAPK1 signaling pathway [122] or the E2F family of transcription factors, which have been linked to neuronal death, neuroinflammation and tissue damage [123]. Several miR-320a-3p targets are related to stress response and pain, including *NR3C1* and *FKBP5*, and are involved in chronic musculoskeletal pain after traumatic events [60].

Additionally, miR-146a-5p has been found to be consistently associated with OA [57,84,89,100,124] in different tissues such as plasma, cartilage and blood cells, thus reflecting intra- and extracellular events and representing a potential candidate systemic biomarker of the processes associated with the disease. Its expression is induced in response to an increase in the levels of inflammatory cytokines, such as tumor necrosis factor α and interleukin (IL)-1β, and to alteration of inflammatory factors such as the Toll-like receptor (TLR) and NF-κB. These processes constitute a negative feedback loop that controls immune activation and may modulate OA-related inflammation in early-stage OA through the miR-146a-5p-mediated downregulation of adaptor/scaffold proteins such as IRAK1 and TRAF6 involved in pro-inflammatory and catabolic pathways, including the IL-1 and TLR signaling pathway, known to positively regulate NF-κB [125]. In later stages, miR-146a-5p levels become low in parallel with degenerative changes and progressive degradation of cartilage [100,126], which is associated with pain. In an animal model of OA-related knee joint pain, a dysregulated expression of miR-146a-5p was also found in sensory neurons of the peripheral and central nervous system, thus suggesting its involvement in both the peripheral (knee) and central pain transmission; moreover, its upregulation in glial cells decreased the expression of pain-related inflammatory cytokines and pain mediators (TNFα, COX-2, iNOS, IL-6, IL-8, RANTS and ion channel, TRPV1) [125].

Many pain conditions, however, remain insufficiently characterized in terms of miRNA expression changes or have not been associated with specific miRNAs, thus requiring further studies to better characterize the molecular (miRNA) fingerprint of pain. For instance, a variety of miRNAs have been intensively researched and investigated in cancers, but the involvement of miRNAs in cancer-related pain is poorly understood.

The bioinformatic analysis of the dysregulated miRNAs showed that the predicted miRNA targets play a role in several cellular functions and cell signaling. The most relevant pathways are involved in signal transduction (e.g., PI3K-Akt signaling, MAPK signaling, Ras signaling, ErbB signaling, Calcium signaling), the endocrine system (e.g., insulin signaling), the immune system (e.g., Toll-like receptor signaling, Leukocyte transendothelial migration, Chemokine signaling) and neuronal functions (e.g., neurotrophin signaling, glutamatergic synapse, GABAergic synapse, cholinergic synapse) that have been associated with pain pathology [127]. For instance, the intracellular signal transduction pathways found here to be associated with miRNA targets are known to play critical roles in pain processes involving neurotransmission, inflammation, neuronal plasticity and hypersensitivity through the modification of receptor/channel phosphorylation or the long-term adaptive regulation of gene transcription and translation [128]. Similarly, tissue infiltration and activation of immune cells at the site of injury, including mast cells, neutrophils, macrophages and T lymphocytes, as well as glial cells, and their release of inflammatory factors, mainly TNF- α, interleukins, interferon- α, prostaglandins, chemokines and leukocyte elastase, are involved in the development and progression of pain [129,130].

A great advancement in pain research and clinical application would be the possibility of discriminating between primary and secondary pain. Besides, though very preliminary, the observation that some miRNAs and related pathways are differentially deregulated in primary and secondary CP would pave the way for confirmatory experimental studies on the potential use of miRNAs for differential diagnosis between these two painful conditions. Our computational analysis suggests that although nearly the same pathways are predicted targets by miRNAs in CPP and CSP, a greater pathways enrichment can be observed in CPP compared with CSP. Moreover, in CPP, several pathways related to neuronal function have also been identified, and most of them are implicated in the transmission and regulation of pain, such as the Neurotrophin signaling pathway [131]; synapse function, e.g., Dopaminergic synapse [132]; Cholinergic synapse; Glutamatergic synapse; Serotonergic synapse; GABAergic synapse [20] and synapse plasticity (e.g., long-term potentiation). Interestingly, pathways related to substance dependence (e.g., amphetamine addiction, morphine addiction, cocaine addiction, alcoholism, nicotine addiction) also emerge in CPP, in accordance with previous propositions that there are anatomical, clinical and neurobiological similarities between chronic pain and addiction, which include overlapping neural circuitry and neuroadaptations including reward deficiency, impaired inhibitory control, compulsive drug seeking and high stress [133,134]. Therefore, miRNAs may represent a proximal cue in the biopsychosocial system of chronic pain and a biological link between chronic pain and neurobiological dependence that deserves further investigations.

This review highlights the use of diversified methodologies in miRNA analysis. Some factors, such as the heterogeneity in the tissue samples and the different extraction and detection techniques, make the comparison among studies hard, suggesting the need for a standardized approach.

The analyzed studies have some limitations. Considering the frequent low differences between miRNA expression in diseased versus healthy subjects, it is important to control some factors such as miRNA pre-processing and normalization experiments, data processing and optimization. In this regard, qRT-PCR is the most widely used method, which can be considered the “gold standard” for the verification of microarrays and next-generation sequencing results due to its greater precision and sensitivity, rapidity and ease of use [135]. In addition, the small number of participants and the lack of result validation in independent cohorts could lead to less coherent results, which is why the works with a limited sample size and those with poor study design should be considered with caution. An important aspect to consider is that, in most studies concerning miRNA profile in CSP, comparisons were made between healthy controls and painful diseased subjects, thus limiting the ability to discern whether miRNAs are pain-specific or are mainly associated with—or modulated by—the disease causing pain. Again, the lacking or insufficient information on gender differences or the impact of age, BMI and medications (including analgesic treatments), which can affect miRNA expression, can also lead to heterogeneity of study results. In many studies analyzed here, no data were reported on the association of deregulated miRNAs with pain symptoms and their evolution in the cohorts studied. Therefore, further longitudinal prospective studies should be conducted because miRNA regulation can change over time during the course of painful disease, and other pathological conditions or confounding factors can overlap over time, which can impact miRNA levels. Importantly, a high throughput analysis of miRNAs was not conducted in the available studies, and only a set of miRNAs or often individual miRNAs were selected for the analysis, so the possibility of missing important miRNAs as potential biomarkers of CP could not be excluded. Information on the role or function of most of miRNA candidates remains incomplete, and, therefore, involvement in the induction and/or maintenance of CP is unknown. Indeed, most studies adopted in silico predictive analysis, so further experimental in vitro or in vivo studies are warranted to validate the targets and determine if the predicted miRNA/mRNA interaction results in significant biological changes.

Notwithstanding limitations and challenges, miRNA research has grown to improve the understanding of the potential role for these molecules in pain physiopathology, eventually enabling this role to be exploited for pain monitoring and management.

## 4. Materials and Methods

### 4.1. Literature Search Strategy

An electronic literature search of MEDLINE/Pubmed, ISI Web of Knowledge, Scopus and Google Scholar was conducted by three separate investigators to retrieve relevant studies concerning the expression of miRNAs in the pathogenesis of CPP and CSP conditions. The following keywords were used: chronic pain, pain syndrome, painful disorders, miRNA, miR, microRNA, pain diagnosis, pain biomarker, pain pathophysiology, chronic primary pain, chronic secondary pain, in conjunction with keywords related to the different CPP and CSP conditions, as specified in Table 1. Additionally, references from selected original studies and reviews were scrutinized for further relevant evidence. The following inclusion criteria guided the search and selection of studies: all types of study design; publication until July 2021; English language; human studies. We excluded studies that used only a microarray for miRNA detection, since, compared to the other molecular approaches, it may give less reliable results due to its low detection sensitivity, reproducibility and dynamic range [136].

### 4.2. Bioinformatic Analysis

To better understand the potential pathophysiological impact of miRNAs, an in silico analysis was performed to reveal their target genes. For the bioinformatic analysis, dysregulated miRNAs were selected among those reported as implicated in CP by at least three independent studies to ensure consistency. For a more robust selection of the potential target genes, we adopted three widely used databases, i.e., TargetScanVert and miRDB, accessed through miRbase [21] and miRWalk, a resource hosting the predicted as well as the experimentally validated miRNA-target interactions [137]. To further improve the reliability of these results, InteractiVenn [138] was used as a tool to retrieve the overlapping target genes, which were subsequently used for pathway enrichment analyses. Cytoscape software (version 3.7.1) was used to construct the miRNA–mRNA regulatory network of miRNA and their target genes [139].

Pathway enrichment analyses were conducted to better understand the functional role of dysregulated miRNAs. GeneTrail2 was used (version 1.6) [140] as a platform to access the Kyoto Encyclopedia of Genes and Genomes (KEGG) database, using the following settings: over-representation analysis; two-sided as null hypothesis (for *p*-value computation); Benjamini-Yekutieli as method to adjust *p*-values; 0.05 as significance level.

Figures were prepared using R packages [141].

## Figures and Tables

**Figure 1 ijms-23-06016-f001:**
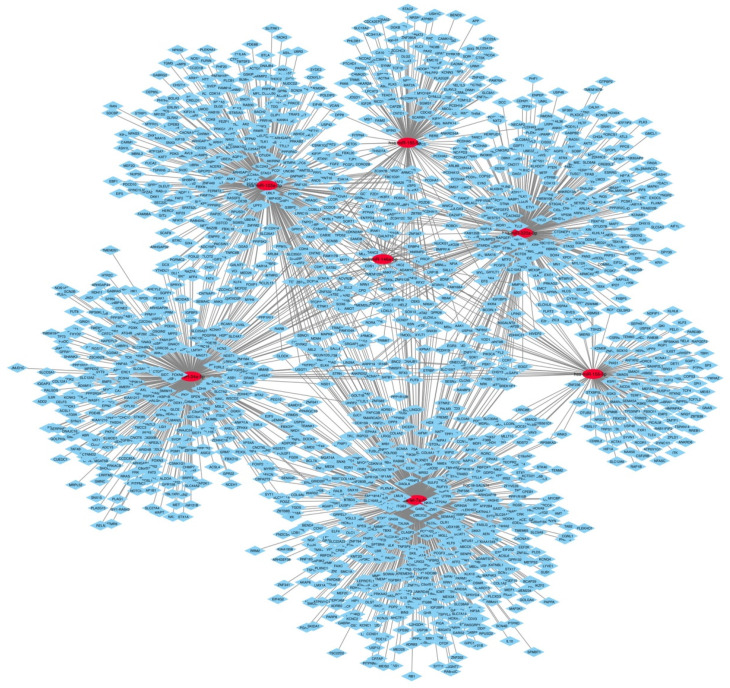
The miRNA–mRNA regulatory network. Red circles represent miRNAs and blue circles represent target genes.

**Figure 2 ijms-23-06016-f002:**
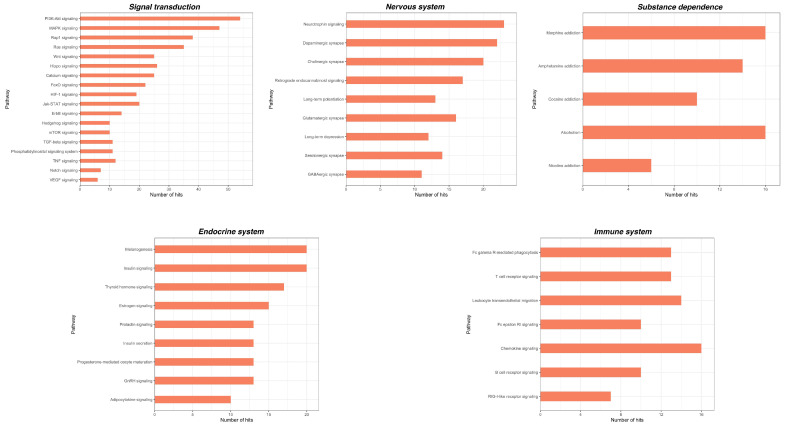
KEGG pathways related to cellular processes significantly associated with miRNA targets in CPP (within each group, pathways are arranged according to their *p*-values, in descending order). Adjusted *p*-values were calculated using Benjamini-Yekutieli method.

**Figure 3 ijms-23-06016-f003:**
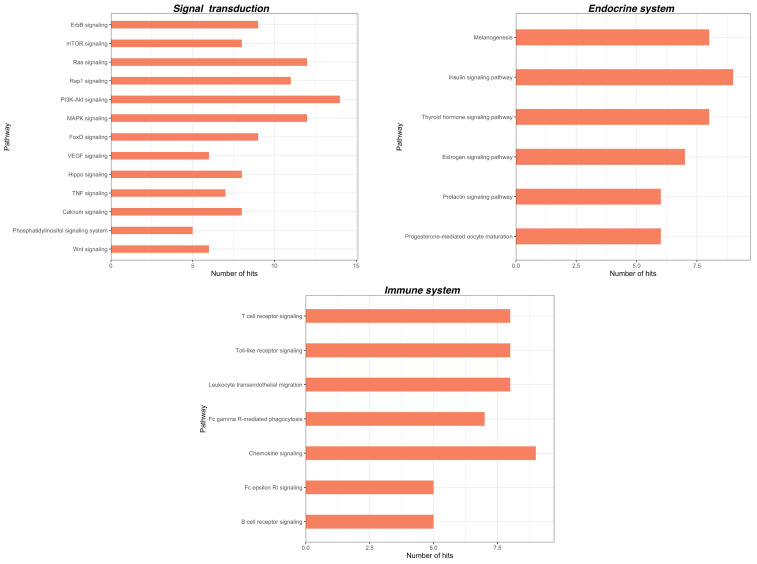
KEGG pathways related to cellular processes significantly associated with miRNA targets in CSP (within each group, pathways are arranged according to their *p*-values, in descending order). Adjusted *p*-values were calculated using Benjamini-Yekutieli method.

**Figure 4 ijms-23-06016-f004:**
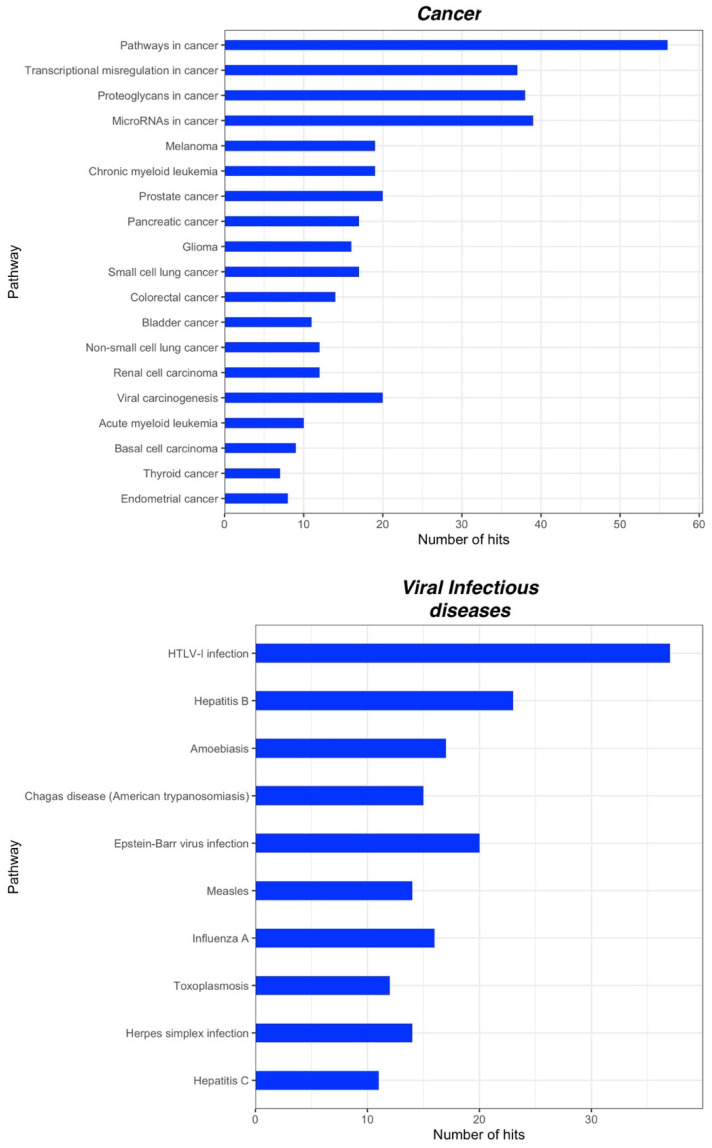
KEGG pathways related to diseases significantly associated with miRNA targets in CPP (within each group, pathways are arranged according to their *p*-values, in descending order). Adjusted *p*-values were calculated using Benjamini-Yekutieli method.

**Figure 5 ijms-23-06016-f005:**
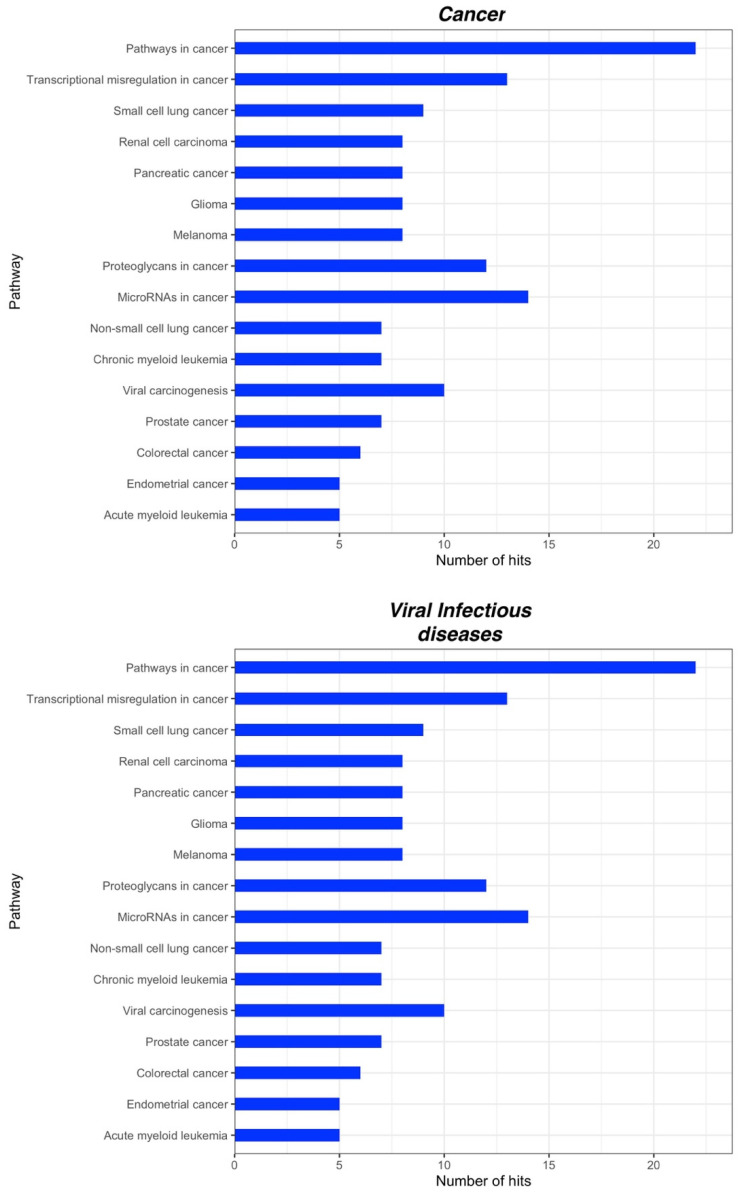
KEGG pathways related to diseases significantly associated with miRNA targets in CSP (within each group, pathways are arranged according to their *p*-values, in descending order). Adjusted *p*-values were calculated using Benjamini-Yekutieli method.

**Table 1 ijms-23-06016-t001:** The IASP classification of chronic pain (modified by [3]).

Chronic Pain	Chronic Primary Pain	Chronic widespread pain Complex regional pain syndrome Chronic primary headache or orofacial pain Chronic primary visceral pain Chronic primary musculoskeletal pain
	Chronic Secondary Pain	Chronic cancer-related pain Chronic postsurgical or posttraumatic pain Chronic neuropathic pain Chronic secondary headache or orofacial pain Chronic secondary visceral pain Chronic secondary musculoskeletal pain

Legend: CM-MO: chronic migraine-overuse of medications; EM: episodic migraine; HC: healthy controls; IBS-C: Irritable bowel syndrome-constipation; IBS-D: Irritable bowel syndrome- diarrhea; IC: interstitial cystitis; IENFD: intraepidermal nerve fiber density; LBP: low back pain; MP: Migraine patients; MPWA: Migraine patients without aura; MPA: migraine patients attack; MPPF: migraine patients pain-free; NR-PE: non-responders plasma exchange; PR: poor responder; qRT-PCR: Quantitative real-time PCR; RE-PE: responder plasma exchange; TN: trigeminal neuralgia. ↓/↑, difference in miRNAs expression, respectively up- and down-expressed.

## Data Availability

Not applicable.

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
