# Peer review of "Expression and Biological Functions of miRNAs in Chronic Pain: A Review on Human Studies"

_ijms, 2022, doi:10.3390/ijms23116016_

Round 1

Reviewer 1 Report

This manuscript attempts to review the literature (published until July 2021) on human studies examining miRNAs in chronic pain pathogenesis (primary and secondary). In addition, they performed target and KEGG analysis for selected miRNAs (found in at least three independent studies) – in total, four for primary chronic pain and three for secondary chronic pain. This manuscript is generally well-written, aside from occasional minor grammatical issues. Please find my few comments/suggestions below:

  • Please consider changing the style of the title and avoid using “chronic human pain”. For example, “Expression and biological functions in chronic pain: a review on human studies.”
  • In the introduction (and methods)part, please rewrite and explain the rationale and approach for miRNA selection for gene targets analysis and analysis itself.
  • Line 685- 687: Authors excluded studies with miRNA detection only by microarray. Please specify here also the approaches that were included/accepted for analysis.
  • Line 691: please avoid jargon and change the word “papers”
  • Figure 1 and 2: Please report statistical tests in the figure captions and, in Figure 2 p values next to the bar or a separate table/supplementary data. Please also indicate the name of the software generating Figures. Improve the quality of the Figures; they are a bit difficult to read. Please generate supplementary data for Figure 1, reporting the raw data in the table for negative and positive interactions.
  • The part of the discussion concerning miRNAs selected for KEGG analysis seems more like a repetition of the results part. Please discuss more the miRNAs chosen concerning available literature in patients with chronic pain that you have used in your analysis.

Author Response

Dear reviewer,

many thanks for the extremely helpful comments and constructive suggestions Please find our point-by-point responses in bold with reference to the modified lines in the manuscript.

  • Please consider changing the style of the title and avoid using “chronic human pain”. For example, “Expression and biological functions in chronic pain: a review on human studies.”

We have modified the title, according with your suggestion (lines 2-3).  

  • In the introduction (and methods)part, please rewrite and explain the rationale and approach for miRNA selection for gene targets analysis and analysis itself.

According with your suggestion, we have better explained this part both in the introduction and method sections (lines 88-153 and 1049-1079).

  • Line 685- 687: Authors excluded studies with miRNA detection only by microarray. Please specify here also the approaches that were included/accepted for analysis.

We have added the inclusion criteria used for the studies selection (lines 1043-1044).

  • Line 691: please avoid jargon and change the word “papers”

We have changed the word “papers” with “studies” within the manuscript (lines 721, 1016).

  • Figure 1 and 2: Please report statistical tests in the figure captions and, in Figure 2 p values next to the bar or a separate table/supplementary data. Please also indicate the name of the software generating Figures. Improve the quality of the Figures; they are a bit difficult to read. Please generate supplementary data for Figure 1, reporting the raw data in the table for negative and positive interactions.

According with your suggestion, we have improved the quality of the Figures. Figure preparation was performed with the use of several R packages (line 1085 and Figures 1-5). Moreover, we have subdivided the old Figures 2 and 3 in more figures in order to be more readable. The statistical test used for the new Figures 2-5 was reported in their captions (lines 778, 794, 805, 810). Finally, we have added supplementary data for Figure 1 (Supplementary materials, Table S1) and supplementary data for Figures 2-5 (Supplementary materials, Tables S2-S3).

  • The part of the discussion concerning miRNAs selected for KEGG analysis seems more like a repetition of the results part. Please discuss more the miRNAs chosen concerning available literature in patients with chronic pain that you have used in your analysis.

Accordingly, we have now discussed more deeply the retrieved miRNAs in relation to chronic pain (lines 851-936).

Reviewer 2 Report

This is a timely and important review of available literature proposing potential role of miRNA in disease pathogenesis and pain process. The use of available data will eventually enable us to understand pain and target it. Besides, the role of miRNA in monitoring of response to drugs can be used. The authors have noted limitations of this review and put forward suggestions for the future research. 

Author Response

Dear reviewer, many thanks for appreciating our work.

Reviewer 3 Report

Excellent article and very informative.  I felt enlightened after reading it.

Author Response

(The authors gave the same response as above.)

Round 2

Reviewer 1 Report

Thank you for the revisions. I have three minor points.

1)     Unfortunately, the legends on the graphs in Figures 2 and 3 are still hardly readable, especially while printed. Please change the quality and

2)     What are the criteria for ordering the targets in Table S1? Is it by p-value or random organization? If p-values, please include p-values in the table. Please organize targets by alphabetic order A-Z if it is a spontaneous organization.

3)     In Table S.2, please change „ , for ” .” in the p-values. 

Author Response

We thank the reviewer for the comments. We have provided the amendments as suggested.

  • Unfortunately, the legends on the graphs in Figures 2 and 3 are still hardly readable, especially while printed. Please change the quality and

We made the quality of the figures as requested by the journal (resolution 300 dpi).

2)     What are the criteria for ordering the targets in Table S1? Is it by p-value or random organization? If p-values, please include p-values in the table. Please organize targets by alphabetic order A-Z if it is a spontaneous organization.

Targets in Table S1 have now been organized in alphabetic order. We have also added it in the title of the table.

3)     In Table S.2, please change „ , for ” .” in the p-values. 

We made the suggested change.